# Does Timing Matter? A Narrative Review of Intermittent Fasting Variants and Their Effects on Bodyweight and Body Composition

**DOI:** 10.3390/nu14235022

**Published:** 2022-11-25

**Authors:** Alan A. Aragon, Brad J. Schoenfeld

**Affiliations:** 1Department of Family and Consumer Sciences, California State University, Northridge, CA 91330, USA; 2Department of Exercise Science and Recreation, CUNY Lehman College, Bronx, NY 10468, USA

**Keywords:** time-restricted eating, nutrient timing, fat mass, body fat, lean mass, fat-free mass

## Abstract

The practice of fasting recently has been purported to have clinical benefits, particularly as an intervention against obesity and its related pathologies. Although a number of different temporal dietary restriction strategies have been employed in practice, they are generally classified under the umbrella term “intermittent fasting” (IF). IF can be stratified into two main categories: (1) intra-weekly fasting (alternate-day fasting/ADF, twice-weekly fasting/TWF) and (2) intra-daily fasting (early time-restricted eating/eTRE and delayed time-restricted eating/dTRE). A growing body of evidence indicates that IF is a viable alternative to daily caloric restriction (DCR), showing effectiveness as a weight loss intervention. This paper narratively reviews the literature on the effects of various commonly used IF strategies on body weight and body composition when compared to traditional DCR approaches, and draws conclusions for their practical application. A specific focus is provided as to the use of IF in combination with regimented exercise programs and the associated effects on fat mass and lean mass.

## 1. Introduction

The age-old practice of fasting is historically rooted in religious traditions. Salient examples include the within-day eating restriction of Ramadan, prolonged fasting in Ancient Greece, and in Old Testament times—typically for spiritual purposes [1]. In addition to cultural and religious significance, the practice of fasting recently has been purported to have clinical benefits, particularly as an intervention against obesity and its related pathologies [2]. Although a number of different temporal dietary restriction strategies have been employed in practice (see Table 1), they are generally classified under the umbrella term “intermittent fasting” (IF).

IF has gained popularity among the general public in recent years. The scientific community has mirrored this interest, as evidenced by a rapid pace of IF-related research publications. IF can be broadly defined as any type of eating pattern involving prolonged periods of fasting (or highly restricted caloric intake) through the course of the day or the week. IF is fundamentally a timing strategy that manipulates the placement and nature of feeding and fasting intervals, which can be stratified into two main categories: (1) intra-weekly fasting (alternate-day fasting/ADF, twice-weekly fasting/TWF) and (2) intra-daily fasting (early time-restricted eating/eTRE and delayed time-restricted eating/dTRE). A growing body of evidence indicates that IF is a viable alternative to daily caloric restriction (DCR), showing similar effectiveness for weight loss [3,4] and alleviating cardiometabolic risk factors [5,6].

The majority of the existing IF literature focuses on the clinical implications of weight loss, with lesser consideration of the effects of IF variants on lean mass and fat mass. Despite the general utility of changes in total body mass as a proxy for changes in fat mass, effects on body composition are ultimately what matters. Changes in lean mass (skeletal muscle in particular) can directly influence resting and active metabolism, as well as functional capacity [7]. Thus, in addition to body weight, the present review explores the effects of various commonly used IF strategies on body composition and draws relevant conclusions for its practical application.

## 2. Intra-Week Fasting

### 2.1. Alternate-Day Fasting

Alternate-day fasting (ADF) is perhaps the most well-studied IF variant [8]. It involves ad libitum 24-h feeding days interspersed with 24 h of fasting. A modified ADF (25% of maintenance needs; approximately 500 kcal consumed on “fasting” days) is the predominant type of ADF in the literature [8]. Investigations of zero-calorie ADF (complete omission of energy intake on fasting days) are scarce but showed similar effectiveness and tolerability compared to DCR [9,10]. Varady and Gabel reported ADF-mediated weight loss ranging from ~3–7% in trials lasting approximately two to six months [11]. A meta-analysis by Elortegui Pascual et al. [12], including 24 randomized controlled trials (RCTs), found that ADF resulted in greater weight loss than TWF and TRE. However, it was noted that the degree of weight loss between the diets was not statistically significant. Effects of ADF on body composition have recently been quantified in a meta-analysis by Cui et al. [13] comprising seven randomized controlled trials (RCTs), where significant decreases in both lean mass and fat mass were reported compared to a control group that followed a habitual diet. Overall, mean differences between the ADF group and the control group in lean mass and fat mass changes were −1.38 and −4.96 kg, respectively. Subgroup analysis showed that in eight-week trials, lean mass and fat mass changes in ADF versus controls were −0.72 and −2.17 kg, respectively; in 12-week trials, these differences were −2.36 and −7.86 kg, respectively.

ADF provides dieters the liberation of not consciously restricting intake for roughly half of the days per week. However, existing long-term data cast doubt on the sustainability of ADF. A 12-month trial by Trepanowski et al. [14] (6 months of weight loss followed by 6 months of maintenance) showed that while ADF yielded similar weight loss and cardiometabolic improvements to DCR, it had slightly greater participant dropout than DCR. A potential criticism of this trial was its assignment of specific caloric targets on feeding and fasting days (weight loss phase: 125 and 25% of daily maintenance needs, respectively; maintenance phase: 150 and 50%, respectively). Assigning ad libitum intake on feeding days could potentially have been less tedious and thus more sustainable. Nevertheless, presenting an ADF option would be beneficial for individualizing programs according to personal preference.

It is worth noting that ADF may have inherent clinical benefits that have the potential to influence body compositional benefits indirectly. Gabel et al. [15] performed a secondary analysis of Trepanowski et al.’s data and found that insulin-resistant subjects had greater improvements in insulin sensitivity via ADF compared to DCR. Importantly, this was observed despite similar body weight decreases and also despite the allowance of 25% of maintenance caloric needs on fasting days. However, caution is warranted against overextrapolating the benefits of ADF (and IF in general). An elegant trial by Templeman et al. [16] compared the following three conditions in an attempt to isolate the effects of fasting from the effects of relative energy balance: (1) “75:75”—linear/daily caloric restriction involving a 25% daily deficit, (2) “0:150”—zero-calorie alternate-day fasting involving a 50% surplus on feeding days (thus making it the deficit equivalent of 75:75), and (3) “0:200”—zero-calorie alternate-day fasting with a 100% surplus on feeding days (thus functioning as a caloric maintenance model of ADF). In sum, this was a meticulous comparison of ADF with linear dietary intake in net hypocaloric conditions as well as eucaloric conditions. Body composition changes were actually superior in DCR (75:75), where 92% of the weight lost was body fat. In contrast, 46% of the weight lost in hypocaloric ADF (0:150) was fat mass, with the majority of reductions in body mass coming from lean mass. No inherent cardiometabolic, hormonal, or gene-specific advantages were seen as a result of fasting, regardless of energy balance. A notable limitation of this trial was its short (3-week) duration, leaving open questions about longer-term effects. It also should be noted that the sample was comprised of lean, healthy subjects, and thus we cannot rule out the possibility that ADF and similar fasting models may have health-related benefits in overweight/obese individuals (and the related pathologies thereof). This is not far-fetched since the primary benefit of IF is the control of excess energy intake. IF, therefore, is likely to have diminishing benefits alongside increases in leanness and fitness. 

The importance of exercise training for achieving and maintaining a healthy body composition cannot be overstated. However, despite the abundance of ADF studies, only a few thus far have examined its combination with exercise [17,18,19] and only two of which assessed body composition. In a 12-week trial on subjects with obesity, Bhutani et al. [17] compared four groups: (1) combination (ADF + endurance exercise), (2) ADF, (3) endurance exercise, or (4) control. The combination of ADF + endurance exercise was the superior performer in terms of lean mass retention and fat mass reduction—as well as favorable shifts in blood lipid profile. A limitation of this study was the use of a single-frequency bioelectrical impedance (BIA) device, which has been shown to overestimate female body fat mass [20]. More recently, Oh et al. [18] compared the eight-week effects of four groups: (1) ADF + exercise, (2) ADF, (3) exercise (combination of aerobic and resistance), and (4) control. All groups showed non-significant decreases in lean mass. ADF + exercise and exercise alone both caused significant reductions in fat mass beyond that of the control group. ADF + exercise was slightly superior to exercise alone, causing 3.3 versus 2.3 kg of fat mass reduction. Limitations shared by both of these studies were a lack of control or accounting of physical activity outside of formal exercise, with the latter study lacking exercise supervision in seven of the eight weeks of intervention. 

### 2.2. Twice-Weekly Fasting

Twice-weekly fasting (TWF) involves two fasting days and five ad libitum feeding days per week. The current body of TWF research involves modified ‘fasting’ days allowing 25% of maintenance caloric intake. The two fasting days can be either on consecutive or non-consecutive days. TWF is less restrictive and thus potentially more realistic compared to ADF by virtue of substantially fewer fasting days throughout the week. Despite the disparity in fasting days, weight loss via TWF is surprisingly comparable to that of ADF, ranging ~4 to 8% in trials spanning 3 to 12 months [21,22,23,24,25,26,27,28,29,30]. In the TWF trials that assessed body composition [21,22,23,25,28,29,30], changes in the lean mass range from −0.7 to −2.2 kg. The majority of these trials show greater lean mass losses in TWF compared to DCR, although these differences tend to lack statistical significance. Similarly, fat mass changes range from −2.5 to −4.7 kg, and these decreases tend to be greater than those via DCR, though not to statistically significant degrees. 

On a related note, Bartholomew et al. [31] recently examined the effects of a once-weekly fasting model with a water-only 24-h fasting day per week on individuals with pre-diabetes and type 2 diabetes (with at least one component of the metabolic syndrome) for 22 weeks. The once-weekly fasting phase was preceded by four weeks of a TWF model involving two zero-calorie days. By the end of the 26-week trial, body weight was minimally affected (−1.7 kg), but improvements were seen in insulin sensitivity and metabolic syndrome scores. Body composition was not assessed, but based on the modest weight loss, the changes in lean mass and fat mass likely lacked practical meaningfulness. Moreover, a high dropout rate (mostly in the control group) and a lack of dietary intake information are limitations to consider. 

As in the case of ADF, studies combining exercise with TWF are scarce. Harvie et al. [21] conducted a four-month trial (three months of weight loss, one month of maintenance) on overweight women. Subjects were assigned a gradual increase in exercise frequency and intensity to reach five 45-min sessions of moderate activity (“walking, strengthening, toning and flexibility exercises”) per week. Three conditions were compared: (1) TWF with fasting days consisting of limited calories (30% of maintenance) and carbohydrates (40 g or less); (2) TWF with fasting days consisting of unlimited protein and fat; (3) DCR at a 25% caloric deficit. Lean mass decreased in all groups, but not to a statistically significant degree. However, fat mass decreases were similar in both TWF variants (−4.3 and −4.1, respectively) compared to DCR (−2.5 kg). It bears mentioning that only 58% of the TWF group planned to continue with the diet beyond six months, as compared to 85% of the DCR group. A recent 16-week trial by Cooke et al. [28] compared the effects of three conditions on subjects with obesity: (1) TWF, (2) supervised sprint interval training (SIT), and (3) a combination of both interventions. The lean mass change was negligible in SIT, while significant decreases occurred in the TWF and combination groups compared to SIT. With the exception of a single time point (8 weeks in the TWF group), protein intakes in all groups ranged from 0.79–0.97 g/kg, explaining the vulnerability to lean mass loss in the dieting groups. Fat mass was unchanged in SIT, while TWF and the combination group’s fat mass changes were approximately −3.5 and −2.5 kg, respectively. 

## 3. Intra-Day Fasting

### 3.1. Time-Restricted Eating (General Findings)

Time-restricted eating (TRE) limits the eating window to 4–10 h (most commonly 8 h), with the remaining 14–20 h in an unfed state. In a similar vein to ADF and TWF models, TRE reduces feeding opportunity, which tends to default to decreased total daily energy intake without purposely tracking or restricting calories during the feeding cycles. The majority of ad libitum/unrestricted TRE studies have resulted in spontaneous decreases in total daily energy intake [32,33,34,35]. For example, LeCheminant et al. [32] demonstrated the simplicity and effectiveness of ad libitum TRE in a two-week cross-over study on healthy young men whose sole assignment was to omit all caloric intake from 7 p.m. to 6 a.m. They were allowed to eat freely outside of that boundary. Energy intake decreased by ~244 kcal/day, leading to significantly decreased body weight compared to the control condition. However, a large, 12-week trial by Lowe et al. [36] compared ad libitum TRE (unrestricted eating from 12 p.m. to 8 p.m.) with a conventional three-meal/day pattern with no time restrictions. No significant fat loss occurred in either group. The only significant change was a loss of lean mass (1.1 kg) in the TRE group. Furthermore, there were no significant within- or between-group differences in measures of glucose control, insulin sensitivity, or blood lipid profile. However, energy and macronutrient intakes were not reported, leaving open questions about between-group differences, especially with protein intake, which could have influenced the differences in lean mass preservation.

Kang et al. [37] recently examined the effects of TRE on anthropometric, metabolic, and fitness parameters in the largest systematic review on TRE to date. Included were 23 longitudinal interventions, 19 of which were RCTs. Eighteen studies reported significant bodyweight reductions averaging 1.9 kg in trials ranging from 1–12 weeks. Weight loss via TRE is roughly 4% on the high end. Ten of the 15 studies that assessed body composition reported a significant reduction in body fat percentage and/or fat mass via TRE. Among these 10 studies, fat mass reductions ranged from 0.4–2.8 kg, with studies ranging from one to three months. Effects on lean mass are mixed but mostly stable. Three of the eleven studies that presented data on lean mass reported a reduction. The majority (seven studies) reported no change. Tinsley et al. [38] conducted a single study in the review that showed a statistically significant increase in lean mass (0.9 kg) during TRE, which involved normal-weight, young women on a structured, progressive resistance training program. Overall, studies involving TRE combined with resistance training tend to show lean mass retention and fat mass reduction comparable to or greater than conventional meal distributions spanning ~12 h [38,39,40], although these findings are not universal. Notably, in Tinsley et al. [35], the lean mass changes in the control group versus the TRE group (+2.3 vs. −0.2 kg, respectively) reflect the possibility that the duration of the TRE group’s feeding window (4 h) was too short to facilitate sufficient nutrient intake (and thus muscle anabolism), despite its allowance of ad libitum feeding. Elite cyclists undergoing TRE have also shown lean mass retention and decreased fat mass [41].

Subsequent TRE + resistance training research has echoed the findings of Kang et al. [37]. An eight-week trial by Kotarsky et al. [42] compared a TRE feeding window (averaging 12 p.m. to 7:30 p.m.) with a control group that maintained their usual feeding pattern (averaging 8 a.m. to 8:30 p.m.). Both groups engaged in aerobic training and resistance training. The TRE group had a greater fat mass reduction than the control group (−3 kg versus −1 kg, respectively). The lean mass remained stable in the TRE group while increasing slightly (+1 kg) in the control group. An intriguing observation is greater fat loss in TRE despite similarly reported decreases in daily energy intake by both groups. It remains uncertain whether this is due to misreporting or to an inherent metabolic advantage of TRE.

Contrary findings to the latter were reported in a four-week trial by Stratton et al. [39], who compared TRE (subjects had the option to eat from 12 p.m. to 8 p.m. or 1 p.m. to 9 p.m.) to a control condition without temporal restrictions. Both groups underwent a periodized, progressive resistance training program. Diets were equated in terms of protein and energy targets. Daily caloric intake assignments were standardized by factoring individual resting energy expenditure (REE) with a moderate physical activity level. A four-compartment model was used to assess body composition. Results showed similar lean mass retention and a decrease in fat mass between groups. Dietary records indicated similar macronutrient intakes between groups. The lack of between-group differences in these parameters—as well as all secondary outcomes, including testosterone, adiponectin, and REE—are notable due to this study’s high degree of rigor. It, therefore, seems likely that any fat loss advantage of TRE is due to lower energy intake (via lower feeding opportunity) rather than some elusive metabolic mechanism inherent to TRE, at least within the short timeframe of this study. 

### 3.2. Early Time-Restricted Eating: A Special Type of TRE?

Early time-restricted eating (eTRE) involves intakes shifted toward the earlier part of the day, from morning to mid or late afternoon. It has been proposed that eTRE patterns align with the body’s circadian clock, thus improving a range of health indexes, whereas late-shifted or delayed time-restricted eating (dTRE) can adversely disrupt circadian rhythm [43]. The most consistent clinical benefits of eTRE compared to dTRE or conventional feeding windows (≥12 h) are improved glucose tolerance and insulin sensitivity, which have been repeatedly demonstrated (largely in sedentary subjects with excess body weight) [44,45]. Nevertheless, when viewing the body of IF research as a whole, it is difficult to uncouple the benefits of feeding temporality from the positive consequences of decreased total energy intake [46].

There is emerging evidence in short-term studies showing eTRE’s therapeutic potential independent of weight loss [47,48,49,50]. A notable example is a five-week crossover study by Sutton et al. [47], who compared the effects of eTRE (6-h feeding window with the final meal consumed by 3 p.m.) with a conventional 12-h feeding window on overweight and obese men. While certain improvements occurred (increased glycemic control/insulin sensitivity, reduced blood pressure, oxidative stress, and evening appetite), there were also adverse effects (increased triglycerides and total cholesterol, increased resting heart rate). While these proof-of-principle findings are interesting, real-world applicability is questionable. Dinner in the majority of the world’s developed populations occurs approximately between 6 p.m. and 9 p.m. To sustain a lifestyle where the final meal (snacks included) occurs significantly before 3 p.m. inevitably would bring about a certain degree of sociocultural dissimilation. 

In addition to the practical and ecological limitations of eTRE, there is considerable evidence challenging the idea that dTRE (especially without excess total energy intake) is detrimental to health. The Muslim holy month of Ramadan involves fasting from food and fluid intake from sunrise to sunset, which is in stark contrast to eTRE. Ramadan research has, by default, provided an interesting counterpoint to the idea that dTRE fosters adverse outcomes. A large systematic review and meta-analysis (91 studies) by Jahrami et al. [51] concluded that Ramadan fasting positively impacts cardiometabolic risk factors. Recent systematic reviews and meta-analyses of Ramadan fasting unanimously report significant weight loss, fat loss, decreased lean mass in non-athletes [52], and better lean mass preservation in exercising individuals [53,54].

Direct, longitudinal comparisons of eTRE versus non-eTRE or dTRE comparators on body weight and body composition are scarce, but the existing literature has yielded mixed but mostly null differences. A recent eight-week trial by Queiroz et al. [55] compared the effects of eTRE (8 a.m. to 4 p.m.), dTRE (12 p.m. to 8 p.m.), and conventional feeding window (8 a.m. to 8 p.m.) on overweight and obese adults. Statistically significant decreases in lean mass and fat mass occurred in all three groups, with no significant differences between groups. However, an unexpectedly higher dropout rate in dTRE, yielded a smaller sample than originally calculated, which may have compromised the power to detect differences. A subsequent 14-week trial by Jamshed et al. [56] compared eTRE (7 a.m. to 3 p.m.) with a conventional feeding pattern (≥12-h window). Weight loss was greater in eTRE. Decreases in fat mass and lean mass were observed in both groups, with no statistically significant between-group differences. In the only long-term (12-month) eTRE trial to date, Liu et al. [57] compared eTRE (8 a.m. to 4 p.m.) with a non-time-restricted control condition in overweight and obese adults in hypocaloric conditions (men were assigned 1500–1800 kcal/day, women were assigned 1200–1500 kcal/day). At 12 months, respective reductions in fat mass (5.9 and 4.5 kg) and lean mass (1.7 and 1.4 kg) in the eTRE and control groups were statistically significant but not different between groups. Furthermore, improvements in glucose, lipids, and insulin sensitivity were not statistically different between groups.

## 4. Executive Summary of the Effects of Intermittent Fasting Variants on Body Weight and Body Composition

IF has achieved mainstream popularity to the degree that the scientific community has taken notice and has been extensively investigating the various forms of IF, which can be categorized as intra-weekly fasting (ADF and TWF) and intra-daily fasting (eTRE and dTRE).When viewing the IF data in aggregate, comparisons to DCR show similar effectiveness for weight/fat loss and improving cardiometabolic health. The underlying mechanism behind these benefits is IF’s facilitation of decreased energy intake and/or the prevention of overeating.ADF is a largely ad libitum (unrestricted) means to default to net weekly hypocaloric conditions. Weight loss from ADF ranges ~3–7% in trials lasting approximately 2 to six months. Mean differences between the ADF group and non-dieting control groups in lean mass and fat mass changes are −1.38 and −4.96 kg, respectively. Dropout rates and poor adherence have been notes of concern in long-term research. In lean, healthy subjects, ADF poses the risk of greater lean mass reduction compared to DCR, with no greater clinical benefits.There is a paucity of research investigating the combined effects of ADF and exercise, but the existing data show that their combination (rather than either alone) is superior for reducing fat mass and preserving lean mass. However, an overly narrow feeding window may compromise lean mass gains during regimented resistance training [35].The TWF method presents a potentially more sustainable alternative to ADF. Weight loss (~4–8% in trials ranging from 3–12 months) is similar to that of ADF despite having more ad libitum feeding days per week. Changes in the lean mass range from −0.7 to −2.2 kg. The majority of these trials show greater lean mass losses in TWF compared to DCR, although these differences tend to lack statistical significance (unlike ADF, whose lean mass losses are significant in the absence of exercise). Moreover, there is a lack of ADF studies that include resistance training, which could at least partially explain the greater prevalence of lean mass loss. Changes in the fat mass range from −2.5 to −4.7 kg.As in the case of ADF, there is a paucity of studies investigating the combined effects of TWF and exercise, but the existing data shows that their combination (rather than either alone) is superior for reducing fat mass and preserving lean mass. It would seem reasonable to employ such a strategy by fasting on non-training days; however, direct research is needed to further explore this hypothesis.TRE presents yet another ad libitum alternative that circumvents the potentially tedious nature of DCR. Bodyweight reductions average 1.9 kg in trials ranging from 1–12 weeks. The high end of weight loss via TRE is roughly 4%, which is significantly lower than that seen in other IF variants. Effects on lean mass are mixed, with the majority of studies showing no significant change. Fat mass reductions range between 0.4–2.8 kg in trials lasting one to three months.An emerging body of investigations on TRE combined with exercise (the majority of which has been resistance training) has consistently shown lean mass retention, with a tendency toward greater fat mass reduction than control diets with conventional feeding windows (even in matched hypocaloric conditions).eTRE has gained scientific interest due to potential clinical benefits that may be independent of weight loss, but the long-term persistence of these effects remains questionable. There has been a general lack of difference in body composition change between eTRE and dTRE (or non-time-restricted hypocaloric controls); all have consistently decreased lean mass and fat mass. This is largely attributable to the lack of studies on eTRE combined with exercise.

## 5. Concluding Perspectives and Practical Applications

It has been fairly well-established that a wide range of meal frequencies and distributions can be effectively utilized to improve body composition [58,59]. Currently, a substantial body of evidence supports IF as a viable dietary approach that performs similarly to conventional, linear dieting for reducing fat mass. The strategy would seem particularly appealing to those who prefer to eat ad libitum, given that IF inherently limits energy intake. It remains unclear if underlying mechanistic factors associated with IF may be more or less favorable for body composition changes in certain individuals as opposed to others based on genetic and/or disease state; this possibility requires further study.

Overall, the IF variants included in this review do not appear to pose greater safety risks compared to conventional/linear dieting. Emerging research has shown TRE to improve subjective ratings of quality of life [60,61]. Systematic reviews and meta-analyses consistently report a lack of serious adverse events across studies [3,62,63]. However, several cautionary notes are warranted in this regard. Cioffi et al. [63] point out that attrition rate and hunger ratings were often higher in IF groups, and the collective evidence does not support “greater ease or acceptability” compared to DCR. Individuals with type 2 diabetes should be cautious about the hypoglycemic potential of IF [64]. Furthermore, IF has been associated with eating disorder symptomology [65], so IF may be risky for individuals struggling with the psychological impacts of food restriction in general. Additionally, a six-month study by Harvie et al. [21] reported significantly longer menstrual cycles in the TWF group compared to the DCR group (mean length of 29.7 vs. 27.4 days, respectively) in young, overweight women. A final note is that internal cues to consume fluids are diminished in the absence of meals [66], elevating the importance of consciously staying sufficiently hydrated on fasting days. Therefore, amidst the benefits, these caveats reinforce the importance of staying vigilant about individual variations in response to IF—or any given dietary approach. 

Although the lack of a universally superior approach might be anticlimactic, the multiple variants of IF provide ample room for individualizing diets according to personal preference, tolerance, and goals. Nevertheless, there is also room for improving/optimizing existing models (see Table 2). A recurrent observation across the IF literature is the loss of lean mass. While exercise (especially resistance training) can at least partially alleviate this, it is also necessary to ensure that daily protein intake is sufficient, at ≥1.6 g/kg [67]. The observational data point to even higher intakes for this purpose. For elite athletes in hypocaloric conditions, Hector and Phillips [68] recommended a range of 1.6–2.4 g/kg, with the severity of the energy deficit and the intensiveness of the training regimen determining where in that range to aim. In ADF or TWF, the modified fasting days involving intakes of ~500–600 kcal are best served by predominating that allotment with high-quality protein sources. Some evidence indicates that spreading protein consumption throughout the day positively influences muscle protein synthesis (MPS) [69,70] and lean mass [71,72]; hence, IF strategies that employ an overly narrow feeding window may be suboptimal in this regard. For example, Tinsley et al. [35] found that a 4/20 feeding/fasting distribution attenuated gains in lean mass in young men undertaking an eight-week resistance training program. Further research is needed to explore the effect of different feeding windows on the anabolic response to protein intake, particularly during regimented resistance training.

As discussed in the present review, substantial evidence supports the effectiveness of IF for body fat reduction and its associated health benefits. However, IF must also be recognized as a double-edged sword to be wielded carefully. Muscle protein exists in a perpetual, dynamic state of ‘turnover’—that is, a continuous cycle of synthesis and breakdown. The net difference between MPS and muscle protein breakdown (MPB) over time determines muscle mass increase, decrease, or stasis. Prolonged fasting directly antagonizes muscle maintenance and growth. The underlying mechanisms of this phenomenon involve decreases in MPS and anabolic signaling activity. To this point, even moderate energy restriction can impede muscle anabolism. Pasiakos et al. [73] found that a mere 20% energy deficit for as few as 10 days decreased MPS by 19% while lowering the phosphorylation of key anabolic signaling molecules, protein kinase B and eukaryotic initiation factor 4E binding protein 1. Vendelbo et al. [74] reported a significant increase in forearm phenylalanine release (indicating net MPB) as a result of a three-day fasting period, during which time a 50% decrease in mTOR phosphorylaytion was also observed. Therefore, fasting cycles are clearly capable of antagonizing or impeding the maintenance or growth of muscle mass to varying degrees, depending on the length and frequency of the fasting cycles. Furthermore, these findings underscore the crucial dependence of muscle on sufficient energy availability for the goal of maintenance and growth. Fulfilling this aim also requires the full spectrum of essential amino acids (attainable through sufficient total daily protein) in conjunction with progressive resistance training. The role of these components in maintaining net muscle protein balance is amplified during energy-restricted conditions [75]. 

It is important to note that the anabolic effect of protein dosing is saturable; that is, a ceiling of MPS stimulation is reached in most populations at ~0.4 g/kg (and as high as ~0.6 g/kg) [76]. Given the anabolic limits of single (protein-rich) meals, IF can compromise muscle growth due to the opportunity costs of its constrained feeding intervals combined with the inherently catabolic nature of its fasting intervals. In this vein, eTRE’s limiting of the final meal to the mid or late afternoon precludes the anticatabolic/muscle-remodeling benefits of pre-sleep protein feeding [77]. This purposeful omission of intake could have negative implications for elderly individuals seeking to preserve lean mass in the face of age-related muscle anabolic resistance. Regardless of population, this lost opportunity for optimizing training recovery/adaptations would be amplified on days involving exercise—particularly resistance training. For the goal of maximizing muscle growth (as opposed to merely preserving it), IF in all of its forms would seem to be suboptimal because it has the potential to compromise net increases in MPS, especially during sustained hypocaloric conditions [78].

## Figures and Tables

**Table 1 nutrients-14-05022-t001:** IF variants and their underlying protocols.

Variant	Subcategory	Protocol
Alternate-Day Fasting (ADF)	Intra-week	Employs ad libitum 24-h feeding days interspersed with 24 h of fasting; can be modified so that 25% of maintenance needs are consumed on “fasting” days
Twice-Weekly Fasting (TWF)	Intra-week	Employs 2 fasting days (consecutive or non-consecutive) and 5 ad libitum feeding days per week; can be modified so that 25% of maintenance needs are consumed on “fasting” days
Early Time-Restricted Eating (eTRE)	Intra-day	Limits the eating window to 4–10 h (most commonly 8 h) with food consumed in the earlier part of the day, with the remaining 14–20 h in an unfed state.
Delayed Time-Restricted Eating (dTRE)	Intra-day	Limits the eating window to 4–10 h (most commonly 8 h) with food consumed in the later part of the day, with the remaining 14–20 h in an unfed state.

**Table 2 nutrients-14-05022-t002:** Improving/optimizing existing IF models with exercise.

Alternate-Day Fasting (ADF) and Twice-Weekly Fasting (TWF)	
Feeding Day	Fasting Day
Total daily protein should be optimized at ≥1.6 g/kg. Coincide feeding days with high-effort or high-volume training days when possible.	For modified fasting (25% of normal maintenance intake; appx 500–600 kcal), choose a protein-dominant meal or snack. Be conscious about staying properly hydrated since cues to consume fluids may be reduced in the absence of meals.
**Time-Restricted Eating (TRE)**	
Consume total daily protein at ≥1.6 g/kg. Tailor the duration of the feeding window (appx. 6–10 h) to individual preference and tolerance. Tailor the placement of the feeding window (early versus delayed) to personal preference, work or lifestyle schedule, and goal (including specific objectives of the training bout). Caveat: goals involving maximal retention (or growth) of muscle mass can benefit from including a pre-sleep protein feeding (~0.4–0.6 g/kg consumed as a final meal/snack before bedtime). This strategy would breach the early-TRE model, but it could be necessary to optimize the aforementioned goals and would especially apply to resistance training days.	

## Data Availability

Not applicable.

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
