# Peer review of "Does Timing Matter? A Narrative Review of Intermittent Fasting Variants and Their Effects on Bodyweight and Body Composition"

_nutrients, 2022, doi:10.3390/nu14235022_

Round 1

Reviewer 1 Report

Firstly, the title and aim of the review as provided at the end of the introduction state that the review is to investigate effects on body composition, yet, many of the conclusions are about body weight or body weight loss, and some of the studies provided within the review do not even measure body composition. The authors need to be clear on the aim of their review and follow this accordingly.  

Secondly, the body of the review largely considers a number of reviews from other authors. Consequently, the need for the review, and the novelty and originality of the work are not clear. The authors need to make the need for their work clear, while also ensuring the work is novel and original. 

Thirdly, some studies are considered in minute detail, while others are barely considered at all. No measures of study quality are included. Without these, either all studies need to be given equal consideration or ideally, measures of study quality will be provided. These measures of quality would then justify the increased consideration of some studies over others, assuming it is the good quality studies that are given more consideration.   

Fourthly, the review has not been conducted in a systematic manner using systematic searches. As such, none of the statements that relate to ‘number of studies’ can be validated. These statements presumably assume that the authors of the cited systematic reviews have done a good job, but they have no evidence of this, and provide no evidence in the paper. Statements relating to ‘number of studies’ need to be reworded, and the limitations of narrative and non-systematic reviews need to be added to the paper.  

Fifthly, the title states interest in timing, but asides from providing some structure very little discussion on timing is included throughout the entire paper. This needs to be rectified.  

Sixthly, the paper ends with a Table 2 of practical recommendations, but these do not stem from the preceding studies nor the preceding discussion. Some of these suggestions have not even been previously mentioned. The conclusions and practical applications have to stem from the preceding discussion – this is imperative. This needs to be rectified. 

Seventhly, to either recommend or not any form of treatment or therapy, researchers need to consider not only the positive effects, but also the potential negative effects. The authors need to include comprehensive consideration of adverse events as part of their review. There is some discussion of tolerance, acceptability and sustainability, but again no evidence is provided. Comprehensive evidence of adverse events and a range of measures related to acceptability, such as quality of life, need to be included. 

Finally, I would like to see considerable detail on potential mechanisms to justify all conclusions. Mechanisms are currently sorely lacking from the paper. 

Author Response

Firstly, the title and aim of the review as provided at the end of the introduction state that the review is to investigate effects on body composition, yet, many of the conclusions are about body weight or body weight loss, and some of the studies provided within the review do not even measure body composition. The authors need to be clear on the aim of their review and follow this accordingly.  

AUTHOR RESPONSE: This is a fair point; as such, the title has been changed to specify “BODYWEIGHT AND BODY COMPOSITION” - as opposed to merely body composition. Also, within the abstract, “bodyweight” has been added to the second-to-last sentence as such: “This paper narratively reviews the literature on the effects of various commonly used IF strategies on bodyweight and body composition…”  Also, the final sentence of the introduction section includes mention of bodyweight as such: “Thus, in addition to bodyweight, the present review explores the effects of various commonly used IF strategies on body composition, and draws relevant conclusions for its practical application. The summary section has also been retitled as follows: “Summary of the effects of intermittent fasting variants on bodyweight and body composition”

Secondly, the body of the review largely considers a number of reviews from other authors. Consequently, the need for the review, and the novelty and originality of the work are not clear. The authors need to make the need for their work clear, while also ensuring the work is novel and original. 

AUTHOR RESPONSE: A large proportion of the publications comprising the discussion within our manuscript are systematic reviews and meta-analyses examining a narrow scope of parameters while lacking interpretation, critique, and application. Thus, we feel that our paper bridges practical gaps left open by the existing literature. For further commentary, please refer to the response to your fourth contention (below); a very similar response applies here.

Thirdly, some studies are considered in minute detail, while others are barely considered at all. No measures of study quality are included. Without these, either all studies need to be given equal consideration or ideally, measures of study quality will be provided. These measures of quality would then justify the increased consideration of some studies over others, assuming it is the good quality studies that are given more consideration.   

AUTHOR RESPONSE: This is a reasonable concern. In order to rectify it, we combed through the individual trials chosen for discussion, and ensured mention of key limitations where appropriate: Templeman et al’s short (3-week) duration, Bhutani et al’s use of single-frequency BIA, Oh et al’s lack of resistance training session supervision, Bartholomew et al’s high dropout rate and lack of dietary intake information, Harvie et al’s disparity in plans to continue 5:2 vs DCR beyond 6 months, Cooke et al’s low protein intakes, Lowe et al’s absence of reporting of energy & macronutrient intakes, Stratton et al’s short timeframe, Queiroz et al’s higher-then-expected dropout rate in dTRE,

Fourthly, the review has not been conducted in a systematic manner using systematic searches. As such, none of the statements that relate to ‘number of studies’ can be validated. These statements presumably assume that the authors of the cited systematic reviews have done a good job, but they have no evidence of this, and provide no evidence in the paper. Statements relating to ‘number of studies’ need to be reworded, and the limitations of narrative and non-systematic reviews need to be added to the paper.  

 AUTHOR RESPONSE: We appreciate your perspective. Prior to writing the review, we discussed whether to opt for a narrative or systematic approach. The benefit of a narrative review is that it allows the authors to provide a broad perspective on a given topic to deepen its overall understanding; on the other hand, a systematic review necessarily is more narrowly focused, and thus limited in the scope of information that can be covered (https://pubmed.ncbi.nlm.nih.gov/29578574/). Given that our objective in this paper was to delve into the nuances of intermittent fasting across a variety of aspects, we ultimately decided that a narrative format would be best to convey the desired information and draw practical conclusions. Ultimately, a narrative review is a viable, and in some instances (as is the case here), more appropriate choice for a review (see for example: https://pubmed.ncbi.nlm.nih.gov/29578574/ and https://pubmed.ncbi.nlm.nih.gov/34305741/). Thus, we believe the narrative approach is a strength here, not a limitation. You make a fair point about rewording instances where we discuss the “number of studies”. However, note that most of these instances were specific to discussion of meta-analyses (which employed systematic searches) that reported these data. In the other instances, we have revised accordingly.  

Fifthly, the title states interest in timing, but asides from providing some structure very little discussion on timing is included throughout the entire paper. This needs to be rectified.  

AUTHOR RESPONSE: To make this more clear, the following has been added before introducing the two main categories of IF (intra-weekly and intra-daily): “IF is fundamentally a timing strategy that manipulates the placement and nature of feeding and fasting intervals, which can be stratified into two main categories:”

Sixthly, the paper ends with a Table 2 of practical recommendations, but these do not stem from the preceding studies nor the preceding discussion. Some of these suggestions have not even been previously mentioned. The conclusions and practical applications have to stem from the preceding discussion – this is imperative. This needs to be rectified. 

 AUTHOR RESPONSE: Agreed. The bullet points in Table 2 regarding diet quality were deleted since they were not discussed leading up to the table, and those points would be somewhat out of place within the article, regardless. Also, the final paragraph was repositioned to appear prior to Table 2 in order for the discussion of the final bullet point (pre-bed protein) to appear prior to its mentioning in Table 2. Also, the following was added to the second paragraph of the concluding section, supporting the bullet point in Table 2 on hydration: “A final note of caution is that internal cues to consume fluids may be reduced in the absence of meals (https://pubmed.ncbi.nlm.nih.gov/3237784/). Thus, consciousness about staying sufficiently hydrated is of elevated importance on fasting days.”

Seventhly, to either recommend or not any form of treatment or therapy, researchers need to consider not only the positive effects, but also the potential negative effects. The authors need to include comprehensive consideration of adverse events as part of their review. There is some discussion of tolerance, acceptability and sustainability, but again no evidence is provided. Comprehensive evidence of adverse events and a range of measures related to acceptability, such as quality of life, need to be included. 

AUTHOR RESPONSE: The following was added to the conclusion section:

“Overall, the IF variants included in this review do not appear to pose greater safety risks compared to conventional/linear dieting. Emerging research has shown TRF to improve subjective ratings of quality of life (PMID: 33911996, PMID: 36235868/. Systematic reviews and meta-analyses consistently report a lack of serious adverse events across studies. (PMID: 29419624, PMID: 30583725, PMID: 32060194). However, several cautionary notes are warranted. Cioffi et al (PMID: 32060194) point out that attrition rate and hunger ratings were often higher in the IF groups, and the collective evidence does not support “greater ease or acceptability” compared to DCR. Furthermore, individuals with type 2 diabetes should be cautious about the hypoglycemic potential of IF (PMID: 29405359). It is also noteworthy that IF has been associated with eating disorder symptomology (PMID: 34191688), so IF may be risky ground for individuals struggling with the psychological impacts of dieting. Furthermore, a 6-month study by Harvie et al (PMID: 20921964) reported significantly longer menstrual cycle in the 5:2 group compared to the DCR group (mean length of 29.7 vs. 27.4 days, respectively) in young, overweight women. A final note is that internal cues to consume fluids are diminished in the absence of meals (PMID: 3237784), elevating the importance of consciously staying sufficiently hydrated on fasting days. Therefore, amidst the benefits, these caveats reinforce the importance of staying vigilant about individual variations in response to IF – or any given dietary approach.”

Finally, I would like to see considerable detail on potential mechanisms to justify all conclusions. Mechanisms are currently sorely lacking from the paper. 

AUTHOR RESPONSE: Agreed that this would enhance the review. The concluding section has been revised to add the following:

“As discussed in the present review, substantial evidence supports the effectiveness of IF for body fat reduction and its associated health benefits. However, IF must also be recognized as a double-edged sword to be wielded carefully. Muscle protein exists in a perpetual, dynamic state of ‘turnover’ – that is, a continuous cycle of synthesis and breakdown. The net difference between MPS and muscle protein breakdown (MPB) over time determines increase, decrease, or stasis of muscle mass. Prolonged fasting directly antagonizes muscle maintenance and growth. The underlying mechanisms of this phenomenon involve decreases in MPS and anabolic signaling activity. Toward this point, complete energy restriction is not even required to impede muscle anabolism. Pasiakos et al (PMID: 20164371) found that a mere 20% energy deficit for as few as 10 days has been shown to decrease MPS by 19% , and also lowered the phosphorylation of key anabolic signaling molecules, protein kinase B and eukaryotic initiation factor 4E binding protein 1. Vendelbo et al (PMID: 25020061) reported a significant increase in forearm phenylalanine release (indicating net MPB) as a result of a 3-day fasting period, during which time a 50% decrease in mTOR phosphorylaytion was also observed. Therefore, fasting cycles are clearly capable of antagonizing or impeding the maintenance or growth of muscle mass to varying degrees, depending on the length and frequency of the fasting cycles. Furthermore, these findings underscore the crucial dependence of muscle on sufficient energy availability for the goal of maintenance and growth. Fulfilling this aim also requires the full spectrum of essential amino acids (attainable through sufficient total daily protein) in conjunction with progressive resistance training. The role of these components in maintaining net muscle protein balance is amplified during energy-restricted conditions (PMID: 24595305). It is important to note that the anabolic effect of protein dosing is saturable; that is, a ceiling of MPS stimulation is reached in most populations at ~0.4 g/kg (and as high as ~0.6 g/kg) (PMID: 25056502).  Given the anabolic limits of single (protein-rich) meals, IF can compromise muscle growth due to the opportunity costs of its constrained feeding intervals combined with the inherently catabolic nature of its fasting intervals. In this vein, eTRF’s limiting of the final meal to the mid- or late-afternoon precludes the anticatabolic/muscle-remodeling benefits of pre-sleep protein feeding 64. This purposeful omission of intake could have negative implications for elderly individuals seeking to preserve lean mass in the face of age-related muscle anabolic resistance. Regardless of population, this lost opportunity for optimizing training recovery/adaptations would be amplified on days involving exercise – particularly, resistance training. For the goal of maximizing muscle growth (as opposed to merely preserving it), IF in all of its forms is suboptimal because it has the potential to compromise net increases in MPS, especially during sustained hypocaloric conditions 65.

Reviewer 2 Report

To the authors; 

This is a very well-written and seemingly well researched paper. I have no comments to provide. 

Author Response

Author Response: We thank the reviewer for your positive feedback and appreciate your taking the time to review.

Reviewer 3 Report

The aim of the paper is to review the existing literature evaluating intermittent fasting protocols and the impact on changes on body composition with or without exercise programming. The paper contributes to the body of literature with a strong summary and analysis of what is currently known about the topic. A strength of the paper is the number of studies available to review in this area and the clearly defined categories of intermittent fasting. In addition, the authors address potential drawbacks of intermittent fasting with certain populations, namely athletes who may desire to change body composition but may negatively impact these goals by applying certain intermittent fasting strategies.

This review is a highly relevant topic for today’s nutrition and dietetics professionals and other health or fitness professionals working with individuals desiring to make health and body composition changes. The topic has received significant media attention. This article can aid health professionals in communicating scientifically sound recommendations to the public.

This review is thorough in the opinion of this reviewer and has incorporated appropriate relevant and current published literature with appropriate supporting references.    

The review is comprehensive and addresses a relevant topic for the public and healthcare professionals.  The authors identified a gap in the knowledge of how intermittent fasting regimens influence changes in body composition. Much of the literature and conversation to date has focused more on weight loss and metabolic changes related to intermittent fasting.

This reviewer is not aware of nor found a similarly published articles. Therefore, this review is likely the first of its kind and highly relevant and of interest to health professionals and the scientific community.

Most cited references are recent publications and are directly relevant to the subject matter and review purpose. The authors do not make excessive reference to their prior work.

The authors draw clear and realistic conclusions from the literature. They address the application of these conclusions in a clear, coherent manner.

For Table 2, recommend adding to the title the words – with exercise (line 340)

Table 2. Improving/optimizing existing IF models with exercise.

The content seems to apply mainly to individuals who follow a regular exercise/training schedule, but may not apply those who are not active and the high protein recommendation may be to high for older adults.

 Overall the review article is thorough and well presented with practical applications to patients/clients and the public.

Author Response

AUTHOR RESPONSE: We thank the reviewer for the positive feedback on our paper, and appreciate you taking the time to review. As per your suggestion, we have added the words “with exercise” to Table 2.
